# Putative Internal Control Genes in Bovine Milk Small Extracellular Vesicles Suitable for Normalization in Quantitative Real Time-Polymerase Chain Reaction

**DOI:** 10.3390/membranes11120933

**Published:** 2021-11-26

**Authors:** Md. Matiur Rahman, Shigeo Takashima, Yuji O. Kamatari, Yassien Badr, Kaori Shimizu, Ayaka Okada, Yasuo Inoshima

**Affiliations:** 1The United Graduate School of Veterinary Sciences, Gifu University, 1-1 Yanagido, Gifu 501-1193, Japan; matiur.vetmed@gmail.com; 2Laboratory of Food and Environmental Hygiene, Cooperative Department of Veterinary Medicine, Gifu University, 1-1 Yanagido, Gifu 501-1193, Japan; yassienbadr1@gmail.com (Y.B.); skaori@gifu-u.ac.jp (K.S.); okadaa@gifu-u.ac.jp (A.O.); 3Department of Medicine, Faculty of Veterinary, Animal and Biomedical Sciences, Sylhet Agricultural University, Sylhet 3100, Bangladesh; 4Division of Genomics Research, Life Science Research Center, Gifu University, 1-1 Yanagido, Gifu 501-1193, Japan; staka@gifu-u.ac.jp; 5Division of Instrumental Analysis, Life Science Research Center, Gifu University, 1-1 Yanagido, Gifu 501-1193, Japan; kamatari@gifu-u.ac.jp; 6Department of Animal Medicine, Faculty of Veterinary Medicine, Damanhour University, Damanhour 22511, Egypt; 7Education and Research Center for Food Animal Health, Gifu University (GeFAH), 1-1 Yanagido, Gifu 501-1193, Japan; 8Joint Graduate School of Veterinary Sciences, Gifu University, 1-1 Yanagido, Gifu 501-1193, Japan

**Keywords:** bovine milk, normalization, small extracellular vesicles, qRT-PCR, putative internal control genes

## Abstract

Bovine milk small extracellular vesicles (sEVs) contain many biologically important molecules, including mRNAs. Quantitative real-time polymerase chain reaction (qRT-PCR) is a widely used method for quantifying mRNA in tissues and cells. However, the use, selection, and stability of suitable putative internal control genes in bovine milk sEVs for normalization in qRT-PCR have not yet been identified. Thus, the aim of the present study was to determine suitable putative internal control genes in milk sEVs for the normalization of qRT-PCR data. Milk sEVs were isolated from six healthy Holstein-Friesian cattle, followed by RNA extraction and cDNA synthesis. In total, 17 mRNAs were selected for investigation and quantification using qRT-PCR; they were further evaluated using geNorm, NormFinder, BestKeeper, and ∆CT algorithms to identify those that were highly stable putative internal control genes in milk sEVs. The final ranking of suitable putative internal control genes was determined using RefFinder. The mRNAs from TUB, ACTB, DGKZ, ETFDH, YWHAZ, STATH, DCAF11, and EGFLAM were detected in milk sEVs from six cattle by qRT-PCR. RefFinder demonstrated that TUB, ETFDH, and ACTB were highly stable in milk sEVs, and thus suitable for normalization of qRT-PCR data. The present study suggests that the use of these genes as putative internal control genes may further enhance the robustness of qRT-PCR in bovine milk sEVs. Since these putative internal control genes apply to healthy bovines, it would be helpful to include that the genes were stable in sEVs under “normal or healthy conditions”.

## 1. Introduction

Extracellular vesicles (EVs) are lipid bilayer nanoparticles found in all bodily fluids, including bovine milk [1]. Bovine milk EVs provide vast biologically important biomolecules, including mRNAs, DNA, lipids, and proteins [1,2]. There have been various classes of EVs, such as exosomes, ectosomes, shedding microvesicles, and apoptotic bodies, according to their size, biogenesis, and release pathways [2]. EVs were isolated by using ultracentrifugation and further filtrated by a 0.22-µm filter, defining one of the EV subtypes so-called “exosomes”. According to the Minimal Information for Studies of Extracellular Vesicles guidelines 2018 (MISEV2018), the use of the term “small EVs” (sEVs) instead of “exosomes” should be suggested [3]. Recent studies have reported that milk sEVs play an important role as intercellular communication mediators between dams and calves and recognized that they are cross-species dispersion elements due to human milk consumption [1,4]. Previous studies revealed the presence of many mRNAs in milk sEVs that play a role in immune modulation and infant immune system growth, development, and maturation [5,6]. As a result, milk sEVs research interests have been continuously growing, mainly for understanding the animals’ physio-pathological status.

Quantitative real-time polymerase chain reaction (qRT-PCR) is a widely used technique for determining and quantifying mRNA [7]. In general, the use of qRT-PCR requires data normalization in relation to the expression of putative internal control genes [8]. The concept behind putative internal control genes selection is that the relative quantity of putative internal control genes should not be affected or regulated by the experimental set-up or physiological conditions [9]. To avoid variance and errors in the results of qRT-PCR normalization, appropriate selection and evaluation of putative internal control genes should be undertaken for each new experimental set-up due to the species and sample variation under study. A number of studies have described the identification of suitable putative internal control genes in different tissues, cells, or cell lines from various species, including humans and animals [10,11,12,13,14]. For example, glyceraldehyde-3-phosphate dehydrogenase (GAPDH) is widely used as a putative internal control gene for human tissues; however, it is inappropriate as a putative internal control gene in various tissues of animal species [15]. Many mRNAs, including ACTB, GAPDH, 18S rRNA, and 28S rRNA, have been used as internal control genes for the normalization of qRT-PCR data [10,11,12,13,14,15]. However, to our knowledge, there has not been any study regarding putative internal control genes in milk sEVs. Therefore, the aim of the present study was to identify suitable putative internal control genes in milk sEVs for qRT-PCR normalization.

The present study investigated a total of 17 candidate putative internal control genes by qRT-PCR, followed by an evaluation of the stability value of eight of these putative internal control genes in milk sEVs using four different algorithms, namely geNorm [16], NormFinder [17], BestKeeper [18], and ΔCT [19]. Furthermore, a comprehensive internal control gene ranking was obtained using the RefFinder software [20]. The study results indicated that TUB, ETFDH, and ACTB are stable putative internal control genes in milk sEVs, suggesting that they could be suitable for normalization in qRT-PCR. The current study also suggests that the selection of optimal putative internal control genes in milk sEVs is a critical aspect that could have a considerable impact on qRT-PCR data analysis.

## 2. Materials and Methods

### 2.1. Sample Collection

Fresh raw milk was collected from six healthy Holstein-Friesian cattle at Yanagido Farm, Gifu University, Japan. After being collected, milk samples were put in sterile jars and transported to the laboratory within 10 min. After that, all milk samples were stored at 4 °C for 10–30 min followed by isolation of milk sEVs was carried out.

### 2.2. Milk sEVs Isolation and Characterization

The milk sEVs isolation and characterization were carried out as described previously [21,22] with slight modifications. Briefly, raw milk was centrifuged at 2000× *g* at 4 °C for 20 min using an A508-C rotor (Kubota, Tokyo, Japan) in a model 7000 centrifuge (Kubota) to remove milk fat globules, somatic cells, and debris. The milk fat was removed by white stick and the defatted milk was poured in a beaker for further process. The defatted milk was preheated at 37 °C for 10 min. For efficient isolation and purification of milk sEVs, acetic acid was added (finally 1%) to the defatted milk. Casein was separated by centrifugation at 5000× *g* for 20 min using a (Kubota) and collection of supernatant milk serum (whey). Whey was filtrated by using 1.0, 0.45, and 0.2 μm pore-size filters (GA-100, C045A047A, and C020A047A, Advantec, Tokyo, Japan). Further, the milk sEVs were concentrated from the whey by ultracentrifugation (UC) at 100,000× *g* at 4 °C for 1 h using a P42A angle rotor (Hitachi Koki, Tokyo, Japan) in a Himac CP80WX ultracentrifuge (Hitachi Koki). The supernatant was discarded and the bottom pellet was resuspended with distilled water (DW) up to 10 mL into a 13PET tube (Hitachi Koki). The UC was carried out again at 100,000× *g* at 4 °C for 1 h using a P40ST swing rotor (Hitachi Koki). Finally, the supernatant was removed and the bottom layer milk sEVs pellet was collected for further use.

For the characterization of milk sEVs biophysically, transmission electron microscopy (TEM) and nanoparticle tracking analysis (NTA) was carried out as described previously [21,22] with slight modifications. In brief, the milk sEVs pellet was diluted 10 times using DW and applied to glow-discharged carbon support films on copper grids. The milk sEVs pellet was stained by using 2% uranyl acetate and dried in a silica chamber. The milk sEVs morphology was observed by an electron microscope, JEM-2100F (JEOL, Tokyo, Japan) at 200 kV. NTA analysis of milk sEVs was carried out by using a NanoSight LM10V-HS, NTA 3.4 instrument (Malvern Panalytical, Malvern, UK) by an entrusted company (Quantum Design Japan, Tokyo, Japan).

### 2.3. RNA Extraction and cDNA Synthesis

Total RNA was extracted from milk sEVs using a Maxwell RSC simplyRNA Tissue Kit (AS1340, Promega, Madison, WI, USA). RNA quality was determined using a 2100 Agilent Bioanalyzer system (Agilent Technologies, Santa Clara, CA, USA). Contaminating DNA was eliminated by treating the samples with DNase I (18068-015, Invitrogen, Carlsbad, CA, USA). cDNA was then synthesized using a PrimeScript RT Master Mix (RR036A, Takara Bio, Kusatsu, Japan) according to the manufacturer’s instructions.

### 2.4. Selection of Internal Control Genes and Primer Design

In the present study, 17 candidate mRNAs were used for selecting stable putative internal control genes in milk sEVs. Most of these candidate mRNAs, corresponding to ACTB, DCAF11, DGKZ, STATH, TUB, ETFDH, EGFLAM, GAPDH, NVL, B2M, ALB, OAZ1, YWHAZ, TBP, and PRKG1 were selected from the raw data of the microarray analysis of our previous study [21]. These genes were chosen based on the low standard deviation (SD) value from the raw data of microarray analysis (data were not shown). The remaining mRNAs such as 18S rRNA [23] and 28S rRNA [24] were selected because previous studies used them as putative internal control genes. After selecting these mRNAs, primers sequences of ACTB, DCAF11, DGKZ, STATH, TUB, ETFDH, EGFLAM, GAPDH, NVL, B2M, ALB, and OAZ1 were designed using Primer BLAST software from the National Center for Biotechnology Information (http://www.ncbi.nlm.nih.gov/tools/primer-blast/, (accessed on 16 November 2020). Furthermore, the sequences of the primers YWHAZ [25], TBP [23], PRKG1 [23], 18S rRNA [23], and 28S rRNA [24] were taken from the previously published papers. The appropriate amplicon sizes were considered at 100–250 bp. Primer information is listed in Table 1 and Appendix A.

### 2.5. qRT-PCR

qRT-PCR was carried out in a total reaction volume of 20 µL including 10 µL of PowerUp SYBR Green Master Mix (Thermo Fisher Scientific, Waltham, MA, USA), 1 µL each of forward and reverse primers (0.5 μM) (Table 1), 2 µL of cDNA, and 6 µL of PCR grade water. qRT-PCR was performed using a Step One Plus thermal cycler (Applied Biosystems, Waltham, MA, USA) in a 96-well optical plate (Applied Biosystems). The following amplification conditions were used: 2 min at 50 °C, 2 min at 95 °C, 40 cycles: 3 s at 95 °C (denaturation) and 30 s at 60 °C (annealing and extension). A dissociation protocol with temperatures of 95 °C for 15 s, 60 °C for 1 min, and 95 °C for 15 s was used to investigate the specificity of the qRT-PCR reaction and the presence of primer dimers. For each mRNA, qRT-PCR was performed in duplicate (technical replicates). The qRT-PCR Excel data for each of the mRNAs were extracted and analyzed further.

### 2.6. Analysis of Internal Control Genes Stability

The stability of the candidate putative internal control gene was analyzed using some of the major computational programs currently available, including geNorm [16], NormFinder [17], BestKeeper [18], and the comparative ∆CT method [19] (Table 1). Using raw non-normalized expression values, the geNorm software [16] was used to determine the candidate putative internal control genes stability values (M). The average pair-wise variation of each putative internal control gene was considered in relation to all, allowing the least stable gene to be eliminated. Following that, the M values were recalculated, resulting in a ranking of the stable genes, with the lower the M value indicating greater gene stability. According to the geNorm software, a stable putative internal control gene should have an average geNorm M value ≤ 1.0 [16]. For determining suitable putative internal control genes, the NormFinder software [17] was used as an alternative algorithm to the geNorm algorithm. Using raw non-normalized data in the form of expression values generated using the comparative CT-method, the NormFinder software directed a model-based approach to determine the expression stability of putative internal control genes. In geNorm and NormFinder software, raw CT values were converted to relative quantities using the 2^−delta CT^ equation, where delta CT = CT sample − CT min (CT sample is the raw CT value and CT min is the least raw CT value) [16,17]. BestKeeper software was calculated the gene expression variation for all individual putative internal control genes based on CT values. BestKeeper software initially estimated the SD and coefficient of variance (CV) following the determination of stability of gene expression for all putative internal control genes. BestKeeper software is a Microsoft Excel-based stability analysis tool that uses raw CT values of the putative internal control genes [18]. The pair-wise correlation analysis was carried out to calculate the correlation between each gene, providing a Pearson correlation coefficient (r), CV%, SD, and a probability (*p*) value to each combination [18]. The highly correlated genes form an index, which is then used to compute the relationship between each candidate’s putative internal control gene and the index. The gene with the highest r is assessed as the most stable putative internal control gene. In the ΔCT software [19], the mean SD was used to assess the stability of the candidate putative internal control gene. A low SD value indicates a stable putative internal control gene and a high SD value indicates a less stable putative internal control gene. Finally, RefFinder [20] was used to rank all the suitable putative internal control genes in milk sEVs by integrating the results from the other four analyses such as geNorm [16], NormFinder [17], BestKeeper [18], and the comparative ∆CT method [19].

## 3. Results

### 3.1. Milk sEVs Isolation and Characterization

TEM analysis revealed the morphology of milk sEVs that exhibited a spherical bilayered shape (Figure 1A). NTA showed that the peak (mode) intensities for the particle size distribution of the milk sEVs was 172.4 (Figure 1B) (representative cattle no. 6). The particle concentration was 3.82 × 10^12^ ± 1.48 × 10^11^ particles/mL (Figure 1B). The results indicated the confirmation of the presence of milk sEVs in the current study.

### 3.2. qRT-PCR

Based on microarray data from our previous study [21] and other published papers [23,24], 17 candidate putative internal control genes in milk sEVs (n = 6) were quantified by qRT-PCR. However, only eight were detected (Table 1; Appendix A). High-abundance putative internal control genes, including TUB, ACTB, and YWHAZ, had average cycle threshold (CT) values ranging from 22 to 28, whereas low-abundance putative internal control genes, including DGKZ, DCAF11, STATH, ETFDH, and EGFLAM, showed average CT values ranging from 29 to 39 (Table 2). The lower CT values reflect the higher mRNA transcript levels i.e., high-abundance genes and higher CT values reflect the lower mRNA transcript i.e., low-abundance genes. The comparatively low CT values of TUB, ACTB, and YWHAZ indicated that they were relatively stable putative internal control genes in milk sEVs for qRT-PCR.

### 3.3. Evaluation of Putative Internal Control Genes Stability

The geNorm analysis [16] showed that ACTB, TUB, and YWHAZ were stable putative internal control genes, whereas DGKZ and STATH were the least stable putative internal control genes in milk sEVs (Figure 2A). The results also showed that the best putative internal control gene combination was ACTB and TUB. According to the NormFinder analysis [17], stable putative internal control genes were TUB, ETFDH, and ACTB, whereas the least stable putative internal control genes were DGKZ and STATH (Figure 2B. The BestKeeper analysis [18] identified TUB, ACTB, and YWHAZ as stable putative internal control genes (Figure 2C and Table 3), whereas STATH and DGKZ were identified as the least stable putative internal control genes in milk sEVs. The stability of the putative internal control genes, according to the ΔCT analysis, is shown in Figure 2D. Among all, TUB, ACTB, and ETFDH were the stable putative internal control genes in milk sEVs, whereas DCAF11 and STATH were the least stable putative internal control genes. The use of different analyses, i.e., geNorm, NormFinder, BestKeeper, and ΔCT, resulted in different stability rankings for the putative internal control genes under study. Therefore, the ranking of the candidate putative internal control genes was further evaluated using RefFinder [20]. The overall final rankings (geometric means or geomean values) are shown in Figure 2E. These final rankings indicated that the stabilities of the eight putative internal control genes in milk sEVs were as follows: TUB > ETFDH > ACTB > EGFLAM > YWHAZ > DGKZ > STATH > DCAF11.

## 4. Discussion

qRT-PCR has been widely used to calculate gene expression levels because of its high sensitivity and specificity. To eliminate non-biological variations, gene quantification analysis involving qRT-PCR requires stringent normalization strategies. Among the several approaches proposed, the use of putative internal control genes is currently the preferred method of normalization [26]. However, the use of incorrect putative internal control genes is known to produce erroneous results [27]. A previous study found that using a single putative internal control gene can also result in gene expression quantification error values (up to 20-fold higher values), implying that multiple putative internal control genes are required for normalization [16]. A large number of studies have been conducted to validate putative internal control genes in many different tissues and cell types [10,11,12,13,14,15]. For example, Lisowski et al. [15] identified ACTB, TBP, YWHAZ, and GAPDH as putative internal control genes in different tissues of bovine origin. In addition, in recent decades, studies of putative internal control genes from sEVs have attracted a lot of interest for their diagnostic and therapeutic potential. However, there is a lack of studies on putative internal control genes in milk sEVs. To the best of our knowledge, this is the first study to identify suitable putative internal control genes in milk sEVs.

In the current study, 17 mRNAs were selected to determine whether they would be useful as appropriate putative internal control genes in milk sEVs for qRT-PCR normalization (Table 1; Appendix A). However, out of such 17 candidates, only eight were detected in milk sEVs by qRT-PCR analysis (Table 1). It is worth noting that widely used putative internal control genes, such as 18S rRNA, 28S rRNA, and GAPDH, were not detected in milk sEVs by qRT-PCR analysis. Further, to identify the stable mRNAs among candidate putative internal control genes in milk sEVs, we used four different analysis methods: geNorm [16], NormFinder [17], BestKeeper [18], and ∆CT [19]. Currently, these are common and widely used algorithms for determining the stability of putative internal control genes in terms of sample origin. The stability rankings of the putative internal control genes in milk sEVs corresponded to TUB, ACTB, and YWHAZ according to the geNorm analysis and to TUB, ETFDH, and ACTB according to the NormFinder analysis. Additionally, the geNorm analysis predicted that TUB and ACTB would be the reliable putative internal control genes pair. Both geNorm and NormFinder analysis recommend a stability value cut-off of 0.15 for identifying the stable putative internal control genes, and all the putative internal control genes from the current study were above this cut-off value. However, in the geNorm algorithm, the most important criterion for evaluating putative internal control genes is the use of a pairwise comparison approach for the determination of gene stability values [16]. In contrast, NormFinder evaluates putative internal control genes using an intra- and inter-group variation approach, avoiding the influence of gene co-regulation [17]. In this approach, ideal putative internal control genes are expected to have stable values, indicated by a low variation in the sample.

According to the BestKeeper analysis, TUB, ACTB, and YWHAZ were stable internal control genes in milk sEVs. In BestKeeper, suitable putative internal control genes are expected to have stable values, as indicated by a low variation among the samples under consideration [18]. Furthermore, in BestKeeper, the Pearson correlation coefficient (r) values are the most important criteria for evaluating the stability of putative internal control genes. This algorithm uses a pairwise correlation analysis for all pairs of candidate putative internal control genes based on the raw CT values and calculates the geometric mean of the best-suited ones. In the ∆CT method, the candidates for stable putative internal control genes in milk sEVs were TUB, ACTB, and ETFDH. In this algorithm, calculations are carried out by comparing the raw CT values obtained in qRT-PCR for the different putative internal control genes under study [19]. Unfortunately, we were unable to identify any putative internal control genes that were consistently stable throughout all four analyses due to the fact that these algorithms use different calculations and methods depending on pairwise comparisons or model-based comparisons, resulting in inconsistencies in the results. To address this limitation, the RefFinder software [20] was used to calculate the stability ranking of the putative internal control genes under study. Our findings suggest that TUB, ETFDH, and ACTB are stable putative internal control genes in milk sEVs and thus are suitable for normalizing qRT-PCR expression data.

Regrettably, the study should concur with previously published papers that have stated that “the ideal internal control genes for qRT-PCR normalization does not exist” [16,26]. There is no easy solution to the question of which algorithm should be used to identify the best putative internal control genes regardless of sample variations [28,29]. It might also be considered that RNA stability and putative internal control genes expression stability may be changed or influenced by the sample collection, pre-processing procedure, method of RNA extraction, sample storage, and time [30,31]. Moreover, using artificial putative internal control genes such as 18S rRNA and 28S rRNA in milk sEVs for normalization in qRT-PCR is a questionable approach.

This is the first study that described the identification of suitable putative internal control genes in milk sEVs for normalizing at qRT-PCR. The main limitation of the study is that we did not validate our data by using a large number of samples at once. The most stable putative internal control genes under certain settings may vary depending on the samples and experimental conditions. Therefore, a detailed validation of candidate putative internal control genes should be carried out according to each particular experimental protocol and design, using a large number of treated and untreated samples.

## 5. Conclusions

The current study attempted to identify suitable putative internal control genes in milk sEVs for the normalization of mRNA expression using qRT-PCR. Our results indicated that TUB, ETFDH, and ACTB are highly stable putative internal control genes in milk sEVs. The results of this study could be useful in developing a quick and effective guideline for selecting appropriate putative internal control genes in milk sEVs to strengthen mRNA normalization in qRT-PCR. The present study also suggests that the use of suitable putative internal control genes in milk sEVs will make it easier to generate reliable, robust, and reproducible results in qRT-PCR for clinical and therapeutic applications. To the best of our knowledge, this is the first study to describe suitable putative internal control genes in milk sEVs.

## Figures and Tables

**Figure 1 membranes-11-00933-f001:**
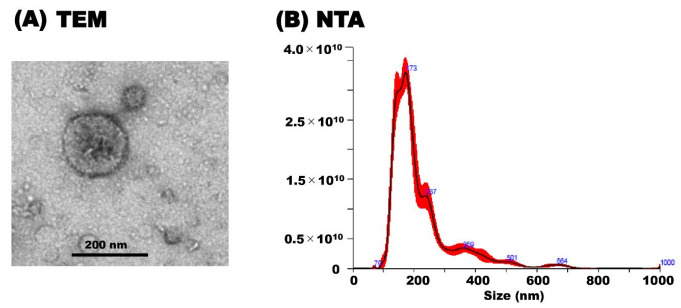
Characterization of milk sEVs. (**A**) Transmission electron microscopy analysis showed the bilayer spherical shape of milk sEVs (Scale bar shows 200 nm in diameter). (**B**) Nanoparticle tracking analysis determined the size distribution of milk sEVs (A representative data from cattle no. 6 was shown (mean peak size < 200 nm in diameter).

**Figure 2 membranes-11-00933-f002:**
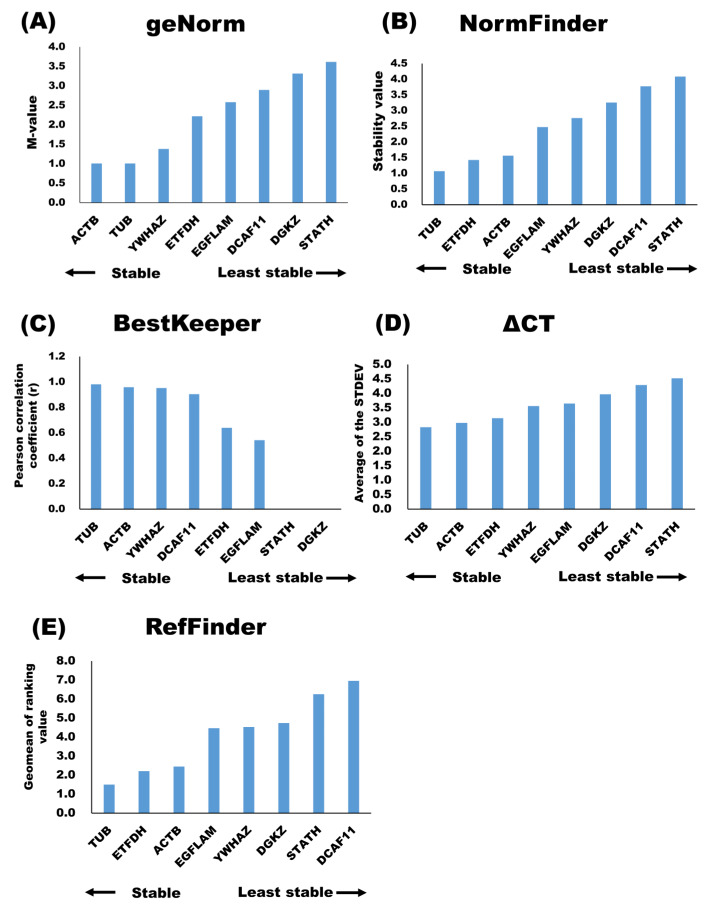
Candidate internal control genes stability in milk sEVs were analyzed by geNorm (**A**), NormFinder (**B**), BestKeeper (**C**), ΔCT (**D**), and geomean of ranking value by RefFinder (**E**).

**Table 1 membranes-11-00933-t001:** Primer sequences used for qRT-PCR of eight putative internal control genes detected in milk sEVs.

Gene Name	Gene Symbol	Primer Sequences	References
Beta-actin	ACTB	F: 5′-GATCTGGCACCACACCTTCTAC-3′	This study
		R: 5′-AGGCATACAGGGACAGCACA-3′	
Statherin precursor	STATH	F: 5′-TACCCAAACCAGCAAGGTGGA-3′	This study
		R: 5′-TGGATACAGCAAGAGGGCAGG-3′	
Electron transfer flavoprotein dehydrogenase	ETFDH	F: 5′-CCAGTGGCTTAGAGGTCCCG-3′	This study
		R: 5′-GGTATACCGGGCAGGCCAAT-3′	
Alpha tubulin	TUB	F: 5′-TGGAACCCACAGTCATTGATGA-3′	This study
		R: 5′-TGATCTCCTTGCCAATGGTGTA-3′	
Diacylglycerol kinase zeta	DGKZ	F: 5′-TCCCGGAGAAAGTGTGCAGC-3′	This study
		R: 5′-GAGCCCGATTCACGGAAGGA-3′	
14-3-3 protein zeta/delta	YWHAZ	F: 5′-GCATCCCACAGACTATTTCC-3′	[25]
		R: 5′-GCAAAGACAATGACAGACCA-3′	
DDB1 and CUL4 associated factor 11	DCAF11	F: 5′-CGCTGAGCAGGCTTTGCTTT-3′	This study
		R: 5′-GAGAGGGCCTGGATGAGCTG-3′	
EGF-like, fibronectin type III, and laminin G domains	EGFLAM	F: 5′-CCGTTTTCTCACTTCGGCCC-3′	This study
		R: 5′-CGAAGGGCCCACACAAGTCT-3′	

sEVs, small extracellular vesicles; qRT-PCR, quantitative real-time polymerase chain reaction; F, forward; R, reverse.

**Table 2 membranes-11-00933-t002:** Cycle threshold values from the qRT-PCR analysis of eight candidate putative internal control genes in milk sEVs.

Sample	TUB	ACTB	YWHAZ	DGKZ	DCAF11	STATH	ETFDH	EGFLAM
1	26.99	29.26	34.63	27.80	33.01	31.49	34.03	36.74
2	21.94	26.24	26.61	28.94	30.63	33.16	32.44	30.60
3	23.17	27.23	28.60	29.54	29.84	31.12	33.09	38.62
4	20.29	23.15	24.63	27.61	26.48	28.92	31.39	34.06
5	25.94	28.49	30.35	30.68	37.10	31.62	31.54	35.42
6	18.62	20.39	22.19	32.52	22.55	36.73	31.51	33.42
CT (av.)	22.83	25.79	27.84	29.51	29.94	32.17	32.33	34.81
STDEV	3.23	3.40	4.40	1.86	5.06	2.61	1.07	2.79
CV	0.14	0.13	0.16	0.06	0.17	0.08	0.03	0.08

sEVs, small extracellular vesicles; qRT-PCR, quantitative real-time polymerase chain reaction; av., average; STDEV, standard deviation; CV, coefficient of variation.

**Table 3 membranes-11-00933-t003:** Repeated pairwise correlation analysis among eight putative internal control genes in milk sEVs and with the BestKeeper index.

Pearson Correlation Coefficient (r)								
	ACTB	TUB	DCAF11	YWHAZ	ETFDH	STATH	EGFLAM	DGKZ
TUB	0.96	-	-	-	-	-	-	-
*p*-value	0.00	-	-	-	-	-	-	-
DCAF11	0.92	0.91	-	-	-	-	-	-
*p*-value	0.01	0.01	-	-	-	-	-	-
YWHAZ	0.94	0.98	0.82	-	-	-	-	-
*p*-value	0.01	0.001	0.05	-	-	-	-	-
ETFDH	0.65	0.62	0.33	0.75	-	-	-	-
*p*-value	0.16	0.19	0.52	0.08	-	-	-	-
STATH	−0.47	−0.40	−0.42	−0.41	−0.15	-	-	-
*p*-value	0.35	0.44	0.41	0.43	0.78	-	-	-
EGFLAM	0.45	0.51	0.28	0.54	0.48	−0.38	-	-
*p*-value	0.37	0.31	0.59	0.27	0.34	0.46	-	-
DGKZ	−0.45	−0.36	−0.28	−0.46	−0.46	0.82	−0.11	-
*p*-value	0.37	0.48	0.60	0.36	0.36	0.05	0.84	-
BestKeeper vs. r	0.96	0.98	0.90	0.95	0.64	0.001	0.54	0.001

## Data Availability

Not applicable.

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
