# Peer review of "Putative Internal Control Genes in Bovine Milk Small Extracellular Vesicles Suitable for Normalization in Quantitative Real Time-Polymerase Chain Reaction"

_membranes, 2021, doi:10.3390/membranes11120933_

Round 1
Reviewer 1 Report
This study reports qRT-PCR studies on bovine milk small extracellular vesicles, with analysis by different algorithms, trying to identify the reference genes. The work looks interesting and the writing is generally good. The simple study with limited data needs improvement. More results are needed to support the conclusion. Below are the comments.
- The main work is about identifying the stable genes among the selected genes. The term “reference genes” may misleading. Genes need to meet several criteria so that they can lay claim to the name (i.e., J Appl Genet. 2013 Nov;54(4):391-406.). It is suggested to revised in both the title and main text.
- Please describe the difference between small extracellular vesicles and the common extracellular vesicles.
- How long were the milk samples kept in 4 ℃ before processing? How to make sure the milk keeps stable? Please comment.
- Please add the detailed description on milk sEVs isolation (what have been done?). Base on the current statement, it is difficult for readers to repeat the work.
- Analysis of RGs stability is one of the most important part of this work. Please describe the detailed methods on the analysis.
- Please show the characterization data of sEVs. In addition, how to confirm the gene are from sEVs but not in free form?
- Please describe how is high-abundance and low-abundance defined?
- In table 2, it is suggested to present the average data with standard deviation and statistical analysis.
- In the evaluation of RGs stability, what is the standard of "highly stable" and "weakly stable". Is the difference between the shown data sufficient for this conclusion?
- Please check figure 1a and correct accordingly.
- When the algorithms used resulted in different ranking, why the Refinder can determine the final ranking? What is the shortcomings of using Refinder?
- It is suggested to combined figure 1 and figure 2, which are showing similar information.
- “The findings suggest that TUB, ETFDH, and ACTB are highly stable RGs in milk sEVs and thus are suitable for normalizing qRT-PCR expression data”. It seems the study just finds some genes with higher stability than other genes selected. How can the conclusion be made? What is the standard for defining if a gene is “stable” and “suitable for normalizing qRT-PCR expression data”?
Author Response
Firstly, thank you very much for your valuable comments and helpful suggestions. Our manuscript has been revised in line by following your instructions and guidance. The text changes in the manuscript are in red color. Our replies to your comments are attached herewith.
- The main work is about identifying the stable genes among the selected genes. The term “reference genes” may misleading. Genes need to meet several criteria so that they can lay claim to the name (i.e., J Appl Genet. 2013 Nov; 54(4): 391-406.). It is suggested to revised in both the title and main text.
Our response
Thank you very much for your kind appreciations and suggestions. According to your suggestions, we have used “Internal control genes” instead of “Reference genes” and revised the title and main text thoroughly.
- Please describe the difference between small extracellular vesicles and the common extracellular vesicles.
Our response
As you commented, we have described about the small extracellular vesicles and the common extracellular vesicles in the main text.
- How long were the milk samples kept in 4℃ before processing? How to make sure the milk keeps stable? Please comment.
Our response
Milk was transported from the farm to the laboratory within 10 min after collection, with a cooling box full of ice used for prevention of protein deterioration in the milk during transport. After reaching to the laboratory, the isolation and purification of milk sEVs was carried out immediately with 10-30 min. Milk sEVs is much more stable in storage condition. A previous study reported that milk sEVs can be stored for prolonged periods without causing significant changes in their physical and biological properties (Munagala et al. 2016, Cancer Lett. 371: 48–61). Moreover, mRNA in milk sEVs are stable even if in gastro enteric acid (Izumi et al. 2012, J. Dairy Sci. 95: 4831–4841).
- Please add the detailed description on milk sEVs isolation (what have been done?). Base on the current statement, it is difficult for readers to repeat the work.
Our response
As you commented, we have added the detailed description of milk sEVs isolation and purification in the main text.
- Analysis of RGs stability is one of the most important part of this work. Please describe the detailed methods on the analysis.
Our response
As you suggested, we have added the detailed description of RGs (changed to internal control genes) stability analysis in the main text.
- Please show the characterization data of sEVs. In addition, how to confirm the gene are from sEVs but not in free form?
Our response
According to your comments, we have briefly described the characterization of milk sEVs in the main text. We purified the sEVs from bovine milk followed by extraction of RNA from milk sEVs, where we showed less amounts of contaminants indicating that contamination of free-mRNAs were also less amounts (Rahman et al. 2019, PLoS One, 14: e0222613) and they were not detectable amounts in microarray analysis (Ishikawa et al. 2020, Viruses, 12: 669). During these procedures, milk sEVs were highly purified and contained the genes encapsulated in milk sEVs.
- Please describe how is high-abundance and low-abundance defined?
Our response
The lower CT values reflect the higher mRNA transcript levels i.e. high-abundance genes and higher CT values reflect the lower mRNA transcript i.e. low-abundance genes. We have added information in the text.
- In table 2, it is suggested to present the average data with standard deviation and statistical analysis
Our response
As you suggested, we have revised the Table 2 accordingly.
- In the evaluation of RGs stability, what is the standard of "highly stable" and "weakly stable". Is the difference between the shown data sufficient for this conclusion?
Our response
This study is the preliminary study of identification of stable RGs (changed to internal control genes) in milk sEVs for normalization in qRT-PCR. Unfortunately, we don’t have any standard for describing the "highly stable" and "weakly stable". Thus, we rephrased the text.
- Please check figure 1a and correct accordingly.
Our response
We have revised the Figure 1 accordingly.
- When the algorithms used resulted in different ranking, why the Refinder can determine the final ranking? What is the shortcomings of using Refinder?
Our response
The present study was unable to identify any RGs (changed to internal control genes) that were consistently stable throughout all four algorithms. This is due to the fact that these algorithms use different calculations and methods, resulting in inconsistencies in the results. To address this limitation, the RefFinder software was used to calculate the stability ranking of the RGs (changed to internal control genes) under study. The RefFinder software can calculate the geometric mean of the RGs (changed to internal control genes) for the overall final ranking, based on the rankings from each program such as geNorm, NormFinder, BestKeeper, and ΔCT. We briefly described these facts in the end of the discussion (Xie et al. 2012, Plant Mol. Biol. 80: 75-84).
- It is suggested to combined figure 1 and figure 2, which are showing similar information.
Our response
According to your suggestion, the Figure 1 and 2 were combined together.
- “The findings suggest that TUB, ETFDH, and ACTB are highly stable RGs in milk sEVs and thus are suitable for normalizing qRT-PCR expression data”. It seems the study just finds some genes with higher stability than other genes selected. How can the conclusion be made? What is the standard for defining if a gene is “stable” and “suitable for normalizing qRT-PCR expression data”?
Our response
This is the first study that described the identification of suitable RGs (changed to internal control genes) in milk sEVs for normalizing at qRT-PCR. The main limitation of this study is that we did not validate our data by using a large number of samples at once. The most stable genes under certain settings may vary depending on the samples and experimental conditions. As a result, selecting the most appropriate RG (changed to internal control genes) for a more accurate result using a large number of treated and untreated samples should be needed. We have added this statements at the end of the discussion.

Reviewer 2 Report
Review of Reference genes in bovine milk small extracellular vesicles ...
Regrettably, the English language and style are not fine or perfect as they should be, so more than a minor spell check is still required to bring your scientific English up to acceptable standards. Example, Line 223 "Regrettably, the study should concurre ..." This reviewer does not recollect concur taking a superfluous 'e'.
Abstract
Expand the abstract a little bit, please. For example, since these reference genes apply to healthy bovines, it would be helpful to include that the genes were stable in extracellular vesicles (EVs) under “normal/healthy conditions”.
Methods
Lines 76-77 – Please indicate the time period stating how long the milk was stored at 4C prior to processing, how long was the transit time, describe the conditions of transit, and any other circumstances that may have influenced or stressed the molecular makeup of the samples.
Line 79 – Please provide brief description(s).
For each of the tested primers, was primer efficiency calculated? Was 60C annealing and extension appropriate for all primers to work satisfactorily at or near 100% efficiency?
Results
Lines 118-119 – More rigour in qualifying the results would add credence to this report. Please consider contrasting or at least comparing your observations with those previously reported in the literature to date. One case in point is, Have the authors confirmed the presence of assayed RG RNA in the sequencing data from bovine milk EVs of other studies? Only two are referenced. If authors chose two studies, why those two and not a more extensive sequencing analysis of publicly available data? Did the referenced studies have healthy bovine conditions, or were disease states involved? Did RNA processing and storage time differ?
Discussion
One major limitation is that the study was conducted under a single control condition and it is unknown how the RG RNA presence will change in cases of mastitis for example, or even different lactation stages, time of day for sample collection, milking.
Strengthen the significance of the report: For example, it might also be worthwhile to mention whether RNA stability and RG expression stability were observed to have changed depending on RNA extraction method, sample storage or raw sample.
Authors should mention that artificial reference genes can be added to EV RNA for qPCR purposes, as described for microRNA in Benmoussa et al. 2017. (https://doi.org/10.1080/20013078.2017.1401897).
Further addition needed: It might be useful to have a brief description of benefits and limitations of an artificial reference gene for assaying an endogenous RG for each individual experiment.
Author Response
Firstly, thank you very much for your valuable comments and helpful suggestions. Our manuscript has been revised in line by following your instructions and guidance. The text changes in the manuscript are in red color. Our replies to your comments are attached herewith.
Regrettably, the English language and style are not fine or perfect as they should be, so more than a minor spell check is still required to bring your scientific English up to acceptable standards. Example, Line 223 "Regrettably, the study should concurre ..." This reviewer does not recollect concur taking a superfluous 'e'.
Our response
Thank you very much for your correction. We have revised the text.
Abstract
Expand the abstract a little bit, please. For example, since these reference genes apply to healthy bovines, it would be helpful to include that the genes were stable in extracellular vesicles (EVs) under “normal/healthy conditions”.
Our response
As you suggested, we have added this text into the abstract and revised it.
Methods
Lines 76-77. Please indicate the time period stating how long the milk was stored at 4°C prior to processing, how long was the transit time, describe the conditions of transit, and any other circumstances that may have influenced or stressed the molecular makeup of the samples.
Our response
According to your comments, we have added the time period into the main text.
Line 79 – Please provide brief description(s). For each of the tested primers, was primer efficiency calculated? Was 60°C annealing and extension appropriate for all primers to
work satisfactorily at or near 100% efficiency?
Our response
We didn’t calculate the primer efficiency test. However, according to the qRT-PCR, 60°C (annealing and extension) was appropriate for the primers.
Results
Lines 118-119 – More rigour in qualifying the results would add credence to this report. Please consider contrasting or at least comparing your observations with those previously reported in the literature to date. One case in point is, Have the authors confirmed the presence of assayed RG RNA in the sequencing data from bovine milk EVs of other studies? Only two are referenced. If authors chose two studies, why those two and not a more extensive sequencing analysis of publicly available data?
Our response
We agreed with your comments that we did not compare the other publicly available RNA sequence, only focusing on our previous microarray data. Unfortunately, the bovine milk sEVs research is not so vast and the isolation and purification, and RNA extraction method from bovine milk is quite different from other cells or tissues. This study is the preliminary study of identification of stable RGs (changed to internal control genes) in milk sEVs for normalization in qRT-PCR. In near future, we have a plan to validate the stable RGs (changed to internal control genes) in milk sEVs in an extensive way using others RNA sequences.
Did the referenced studies have healthy bovine conditions, or were disease states involved? Did RNA processing and storage time differ?
Our response
In our previous study, for microarray analysis, milk sEVs were collected from bovine leukemia virus-infected and uninfected (healthy) cattle (Ishikawa et al. 2020, Viruses, 12: 669). Both of the previous and present studies followed same protocol and RNA isolation kit namely, Maxwell RSC simplyRNA Tissue Kit (AS1340, Promega, Madison, WI, USA) for RNA extraction from milk sEVs. There was no variation of the storage time of the milk sEVs.
Discussion
One major limitation is that the study was conducted under a single control condition and it is unknown how the RG RNA presence will change in cases of mastitis for example, or even different lactation stages, time of day for sample collection, milking.
Our response
We agreed with your statements that the main drawback of our study is that we did not validate our data by using a large number of treated and untreated samples.
Strengthen the significance of the report: For example, it might also be worthwhile to mention whether RNA stability and RG expression stability were observed to have changed depending on RNA extraction method, sample storage or raw sample.
Our response
Thank you very much for your suggestions. We added the explanation into the main text with slight modification.
Authors should mention that artificial reference genes can be added to EV RNA for qPCR purposes, as described for microRNA in Benmoussa et al. 2017. (https://doi.org/10.1080/20013078.2017.1401897).
Further addition needed: It might be useful to have a brief description of benefits and limitations of an artificial reference gene for assaying an endogenous RG for each individual experiment.
Our response
Two mRNA genes such as 18S rRNA and 28S rRNA used in qRT-PCR that were not included in our previous microarray analysis (Ishikawa et al. 2020, Viruses, 12: 669), however, they were used as a RGs (changed to internal control genes) in previously published papers (Zhao et al. 206, Korean J. Parasitol. 54, 39-46; Gou et al. et al. 2013, Korean J. Parasitol. 51, 511-517). These two mRNAs were not detected in milk sEVs by qRT-PCR, indicating that artificial RGs (changed to internal control genes) in milk sEVs were used to normalize in qRT-PCR is a questionable approach. We have revised the text.

Reviewer 3 Report
In the paper entitled “Reference genes in bovine milk small extracellular vesicles (sEVs) suitable for normalization in quantitative real time-polymerase 3 chain reaction” the authors analyzed by a bioinformatic approach the stability of mRNA detected in bovine milk sEVs in order to highlight those usable as reference genes (RGs). Since the appropriate selection of RGs for having valid results of qRT-PCR normalization in sEVs is an open question, the approach and results presented offer interesting insights.
The paper has a simple structure but well focused on the aim to suggest which mRNAs can be considered the most stable in the EVs. However, there are some aspects not too much clear which need to be improved as suggested below,
- The description of the method used for the isolation of bovine milk sEVs, even if briefly, has to be reported. To indicate only the references is not enough.
- I suggest to change the title of the table 1 from “Primer sequences used for qRT-PCR of eight RG candidates in milk sEVs” in “Primer sequences used for qRT-PCR of eight RG detected in milk sEVs”.
- I suggest to change the name of y-axis in the graph in Figure 1a from “stability value” to “M-value”. Moreover, in the same graph, it should be better to represent ACTB and TUB separately, even if they have the same M-value.
- The figure 1c and Table 2 are not too much clear. What the values reported in the y-axis are? Are the values reported in the last raw of the Table 3? If yes, the name of the axis should be the same reported in the table. Anyway, the authors should explain better how the BestKeeper index was calculated since it is unclear. In the text the authors wrote: “The BestKeeper algorithm calculates the stability of candidate RGs based on Pearson correlation coefficient (r) and standard deviation (SD) values”; however, in the table 3 no SD values are reported but p values. Moreover, they wrote: “The RGs with the lowest r values were considered to be highly stable. BestKeeper identified TUB, ACTB, and YWHAZ as highly stable RGs (Figure 1c and Table 3), whereas STATH and DGKZ 153 were identified as weakly stable RGs in milk sEVs”. What r values do the authors refer to? In the Figure 1c, TUB, ACTB, and YWHAZ seem have the higher values.
- The number of analyzed samples is too low (only six). The presented data could have more impact increasing this number. If not, the authors should present the study as preliminary and well discuss the limitation of the presented data.
Author Response
Firstly, thank you very much for your valuable comments and helpful suggestions. Our manuscript has been revised in line by following your instructions and guidance. The text changes in the manuscript are in red color. Our replies to your comments are listed below.
The description of the method used for the isolation of bovine milk sEVs, even if briefly, has to be reported. To indicate only the references is not enough.
Our response
Thank you very much for your comments. We have briefly described the isolation
procedure of milk sEVs into the main text.
I suggest to change the title of the table 1 from “Primer sequences used for qRT-PCR of eight RG candidates in milk sEVs” in “Primer sequences used for qRT-PCR of eight RG detected in milk sEVs”.
Our response
As you suggested, we have revised the title of the Table 1.
I suggest to change the name of y-axis in the graph in Figure 1a from “stability value” to “M-value”. Moreover, in the same graph, it should be better to represent ACTB and TUB separately, even if they have the same M-value.
Our response
According to your suggestions, we have revised the Figure 2A (changed from
Figure 1A).
The figure 1c and Table 2 are not too much clear. What the values reported in the y-axis are? Are the values reported in the last raw of the Table 3? If yes, the name of the axis should be the same reported in the table. Anyway, the authors should explain better how the BestKeeper index was calculated since it is unclear. In the text the authors wrote: “The BestKeeper algorithm calculates the stability of candidate RGs based on Pearson correlation coefficient (r) and standard deviation (SD) values”; however, in the table 3 no SD values are reported but p values. Moreover, they wrote: “The RGs with the lowest r values were considered to be highly stable. BestKeeper identified TUB, ACTB, and YWHAZ as highly stable RGs (Figure 1c and Table 3), whereas STATH and DGKZ were identified as weakly stable RGs in milk sEVs”. What r values do the authors refer to? In the Figure 1c, TUB, ACTB, and YWHAZ seem have the higher
values.
Our response
Thank you very much for your observations and suggestions, we have added brief description of all analysis such as geNorm, NormFinder, BestKeeper, ∆CT, and RefFinder into the materials and methods, accordingly. We have revised the Figure 2C and text, according to the Pearson correlation coefficient (r) by BestKeeper analysis. We have also added standard deviation (STDEV) and coefficient of variation (CV) of the cycle threshold values of the internal control genes and revised the Table 2.
The number of analyzed samples is too low (only six). The presented data could have more impact increasing this number. If not, the authors should present the study as preliminary and well discuss the limitation of the presented data.
Our response
We agreed with your statements that the main drawback of our study is that our
analyzed samples were too low. We have added this statement into the discussion.

Round 2
Reviewer 3 Report
The authors have resolved the weaknesses of the first version of the paper and now the most important points have been clarified or well fixed.
In my opinion, the article in its final form can be accepted for publication.
Author Response
Thank you very much for your great appreciation and consideration.
